# Deep Learning-Based Grimace Scoring Is Comparable to Human Scoring in a Mouse Migraine Model

**DOI:** 10.3390/jpm12060851

**Published:** 2022-05-24

**Authors:** Chih-Yi Chiang, Yueh-Peng Chen, Hung-Ruei Tzeng, Man-Hsin Chang, Lih-Chu Chiou, Yu-Cheng Pei

**Affiliations:** 1Department of Physical Medicine and Rehabilitation, Chang Gung Memorial Hospital at Linkou, Taoyuan 33305, Taiwan; t030405@gmail.com; 2Center for Artificial Intelligence in Medicine, Chang Gung Memorial Hospital at Linkou, Taoyuan 33305, Taiwan; yuepengc@gmail.com; 3Master of Science Degree Program in Innovation for Smart Medicine, College of Management, Chang Gung University, Taoyuan 33302, Taiwan; 4Graduate Institute of Pharmacology, College of Medicine, National Taiwan University, Taipei 10617, Taiwan; rogertzeng8@gmail.com (H.-R.T.); r07443004@ntu.edu.tw (M.-H.C.); 5Graduate Institute of Brain and Mind Sciences, College of Medicine, National Taiwan University, Taipei 10617, Taiwan; 6Graduate Institute of Acupuncture Science, China Medical University, Taichung 40447, Taiwan; 7School of Medicine, Chang Gung University, Taoyuan 33302, Taiwan; 8Center of Vascularized Tissue Allograft, Chang Gung Memorial Hospital at Linkou, Taoyuan 33305, Taiwan

**Keywords:** mouse grimace scale, deep machine learning, spontaneous pain, migraine animal model, facial expression

## Abstract

Pain assessment is essential for preclinical and clinical studies on pain. The mouse grimace scale (MGS), consisting of five grimace action units, is a reliable measurement of spontaneous pain in mice. However, MGS scoring is labor-intensive and time-consuming. Deep learning can be applied for the automatic assessment of spontaneous pain. We developed a deep learning model, the DeepMGS, that automatically crops mouse face images, predicts action unit scores and total scores on the MGS, and finally infers whether pain exists. We then compared the performance of DeepMGS with that of experienced and apprentice human scorers. The DeepMGS achieved an accuracy of 70–90% in identifying the five action units of the MGS, and its performance (correlation coefficient = 0.83) highly correlated with that of an experienced human scorer in total MGS scores. In classifying pain and no pain conditions, the DeepMGS is comparable to the experienced human scorer and superior to the apprentice human scorers. Heatmaps generated by gradient-weighted class activation mapping indicate that the DeepMGS accurately focuses on MGS-relevant areas in mouse face images. These findings support that the DeepMGS can be applied for quantifying spontaneous pain in mice, implying its potential application for predicting other painful conditions from facial images.

## 1. Introduction

Pain is an unpleasant emotional and sensory experience associated with actual or potential tissue damage, according to the definition by the International Association for the Study of Pain [1]. The effects and costs of pain are substantial as it could further cause depression [2,3], sleep disturbance [2,3], anxiety [4], negatively affect the quality of life [5], and impose a considerable economic burden on patients, health services, and societies [6]. Headache is a highly common pain disorder, especially migraine.

Migraine is a neurological disorder with symptoms including not only severe headache attacks but also generally associated with nausea and/or light, sound, tactile, and/or hypersensitivity [7]. It is one of the most painful and disabling neurological disorders and has an overall prevalence of approximately 16% in the United States [8,9]. The socio-economic burden inflicted by migraine is insurmountable, as it negatively affects the well-being and productivity of active labor forces [10]. Animal models of migraine, including the repeated nitroglycerin (NTG) model, display both cephalic nociceptive responses and paw allodynia [11]. NTG, a nitric oxide donor, activates nociceptors in the trigeminovascular system (TGVS) and thus triggers migraine attacks [12]. Our understanding of migraine pathophysiology is chiefly based on mouse models, and thus spontaneous pain assessment in mice is receiving increasing emphasis [13].

The mouse grimace scale (MGS), a standardized behavioral coding system, was reported to be able to quantify spontaneous pain in mice with a high accuracy of 81% and has a high inter-rater reliability with an intraclass correlation coefficient of 0.90 [14]. It contains five action units, namely orbital tightening, nose bulge, cheek bulge, ear position, and whisker change, with each being scored as 0, 1, or 2. The total MGS score is the sum of the scores of these five action units. MGS scoring by human raters is time-consuming as the scorers must visually analyze numerous animal images. Furthermore, human-annotated scoring has several limitations, such as the subjectivity and inconsistent application of scoring criteria among scorers, leading to the difficulty in ensuring high-quality scoring.

Some of these shortcomings of human-annotated scoring may be resolved using machine learning, which can learn the algorithm from big data and automatically detect and classify further data. Machine learning’s application in pain scoring has been growing rapidly. For instance, machine learning has been used to estimate the intensity of neonate pain, and its estimation is highly correlated with the scores evaluated by human examiners, showing its potential for automated pain monitoring [15,16]. Besides, convo=lutional neural network models have been applied to assess pain by facial expression in critically ill patients [17]. In animal research, studies have used software that can automatically select the images suitable for MGS analysis [18,19]. For automated MGS scoring, Tuttle et al. [20] developed a machine learning model to automatically yield total MGS scores in mice to access laparotomy-evoked pain and the effect of pain relief, and the scores are highly correlated with those yielded by human scorers. However, they detected pain in a binary manner, i.e., pain or no pain status. To our best knowledge, there has been no study that applied deep learning techniques for automatically predicting total or five action unit MGS scores.

In the present study, our purpose was to develop a deep learning model, the DeepMGS, that automatically estimates the MGS score and reduces the labor and time costs. We used a mouse migraine model induced by repeated and intermittent injections of NTG and a control group injected with saline [21]. The migraine-like painful facial expressions in mice were video-recorded and scored using the MGS [22]. Mouse facial images were classified as the NTG condition, saline condition, and preinjection condition. The performance of the DeepMGS and human-annotated scores in inferring mouse pain were compared. The DeepMGS performed well in scoring the five MGS action units and the total MGS score. The ability of the DeepMGS to classify NTG and saline conditions was comparable to that of an experienced human scorer and superior to that of apprentice scorers, suggesting a promising utility of the DeepMGS in preclinical pain research and potential application to migraine assessment in neonates and critically ill patients.

## 2. Materials and Methods

### 2.1. Animals

All the study data were obtained from our previous study [22], where all animal experiments were approved by the Institutional Animal Care and Use Committee of National Taiwan University, College of Medicine, Taipei, Taiwan and were consistent with the national guidelines. Male mice (ICR strain, 8–10 weeks old, 30–35 g) were used in the experiments. The animals were purchased from BioLASCO (Taipei, Taiwan) and held in 27.5 × 15.5 × 18.5 cm^3^ cages (5 mice/cage) with food and water ad libitum. The cages were placed in a temperature-controlled (23 °C) holding room with a 12 h light/dark cycle (light on at 08:00).

### 2.2. Behavioral Observation

On the day of the experiment, mice in their home cages were moved to the behavioral room and acclimated there for 1 h before testing. After acclimatization, one mouse was placed in each of the four cubicle chambers, which were custom-built in a four-cubicle array (each measuring 7 × 8 × 14 cm^3^). The walls on the back and lateral sides were made of stainless steel and that on the front side was made of transparent Plexiglas. This arrangement encouraged the mouse to look toward the transparent front wall, where a high-resolution (1920 × 1080) digital video camera was placed 0.25 m away and was more likely to capture facial expressions.

In total, the images taken from 12 mice were analyzed in this study. The mice were evenly randomized into migraine and control groups. The migraine group was intraperitoneally injected with NTG solution (10 mg/kg) (Millisrol injection, Nippon Kayaku, Tokyo, Japan), whereas the control group received a saline injection of the same volume. NTG and saline were injected once every other day for five sessions, that is, on Days 1, 3, 5, 7, and 9.

On the injection days, mice were videotaped 10 min before (preinjection period) and 30–60 min after NTG and saline injections. The images for analysis were collected by snapshots and saved in the portable network graphic format in a lossless manner. The images were captured once every 2 min. In total, 1504 images were collected; 223 images sampled before injection with either saline or NTG were grouped as the preinjection condition, and 652 and 619 images were sampled after injection of saline and NTG, respectively.

### 2.3. Human Scoring of MGS and Image Processing

The MGS was used to score painful facial expressions of mice into five action units (Figure 1), each scored as 0, 1, or 2 by human scorers [14]. A score of “0” indicates the scorer had high confidence that the action unit was absent, “1” indicates high confidence of a moderate appearance of the action unit or equivocation over its presence or absence, and “2” indicates high confidence of the marked appearance of the action unit. The total MGS score is the sum of the five action unit scores.

The sequence of images of the three conditions was randomized so that the human scorers were blinded to the image ID. Each image was scored by four human scorers. Among the four scorers, the MGS scores provided by a scorer with 3 years of experience were used as the ground truth and those provided by the other three apprentice scorers were used for comparing DeepMGS performance. For data augmentation, before each training iteration, images were randomly rotated at an angle between −20° and 20° and horizontally flipped with a 50% probability. Image input size was resized to 224 × 224 pixels (bilinear interpolation) with three color channels (8 bits/channel). The aforementioned image preprocessing was performed using the Python Imaging Library.

### 2.4. DeepMGS Development

Among the full dataset of 1504 images, 1127 (75%) were assigned for training, 76 (5%) for validation, and 301 (20%) for testing. The composition ratio of the images of preinjection, saline, and NTG conditions was maintained among the training, validation, and testing datasets (Table 1).

Each image was manually annotated with its scores defined in the five action units (Figure 2a). Because the number of images for each score was not balanced, we adopted an oversampling technique to avoid overweighting a specific score. The DeepMGS consists of five models, namely orbital tightening, nose bulge, cheek bulge, ear position, and whisker change classification models; each model was used for the corresponding MGS action unit. Each model was independently trained, validated, and tested using the aforementioned image sets (Table 1 and Figure 2a). Five-fold cross-validation was performed to train the model. The original dataset was split into five subsets with the same number of images. Each image was randomly assigned once to one of the five subsets. The process of model training was repeated five times with each of the five subsets used once as the testing data. The validation sets were used for monitoring the training losses and early stopping during the training. The results were combined over the five testing subsets to give estimates of the model’s predictive performance. The five models would yield five predicted action unit scores for each image (Figure 2b). The predicted total MGS score was the sum of these predicted action unit scores (Figure 2c). The five-fold validation loss and training accuracy are presented (Figure 2d).

We employed the ResNet18, a convolutional neural network architecture, to predict the scores of five action units. Cross-entropy loss was used as the loss function when optimizing the classification model. The deep learning algorithms in this study were developed using the Ubuntu 18.04 system with NVIDIA 1080Ti GPU 11 GB VRAM. Each training session had 25 epochs, a batch size of 25, and a learning rate of 1 × 10^−4^ with a 20% dropping factor. The models were trained using the Stochastic Gradient Descent with a Momentum (SGDM) optimizer. The SGDM can update the network parameters (weights and biases) to minimize the loss function. The momentum term in the SGDM can reduce oscillations of the path towards the optimum. The models stop training if the training loss did not decrease for four consecutive epochs. The training scripts utilized PyTorch v1.8.0 library and were written in Python v3.8.

### 2.5. Statistics

The correlation between the DeepMGS and ground truth was determined using Pearson’s correlation, and the linear fit was examined through univariate linear regression. The accuracy, sensitivity, specificity, precision, and F1 score analyses were performed using the following equations:$$\mathrm{Accuracy}=\frac{true\;positive+true\;negative}{all\;images\;}$$
$$\mathrm{Sensitivity}=\frac{true\;positive}{true\;positive+false\;negative}$$
$$\mathrm{Specificity}=\frac{true\;negative}{true\;negative+false\;positive}$$
$$\mathrm{Precision}=\frac{true\;positive}{true\;positive+false\;positive}$$
$$F1\;\mathrm{score}=\frac{2*true\;positive}{2*true\;positive+false\;positive+false\;negative}$$
where true positive and true negative are the numbers of images correctly predicted and correctly rejected, respectively. Conversely, false positive and false negative represent the numbers of images incorrectly predicted and rejected in the task, respectively. To obtain the confidence intervals (CIs) of accuracy, sensitivity, specificity, precision, and F1 score, the Clopper–Pearson method was used. To test whether the area under the receiver operating characteristic curves (AUROCs) and the area under the precision-recall curves (AUPRCs) of the DeepMGS when performing classification were significantly higher than the chance level (null hypothesis), bootstrapped resampling was performed 1000 times. The confidence levels of the AUROCs and AUPRCs were determined from the 5th and 95th quantile values of the 1000 bootstrap estimates.

## 3. Results

### 3.1. DeepMGS Achieves High Accuracy in Scoring Individual Action Units

Table 2 lists the performance of each of the five action unit models in terms of accuracy, sensitivity, specificity, precision, and F1 score for the images with scores of 0, 1, or 2. Accuracy was the highest in orbital tightening (86% in score 0, 81% in score 1, and 88% in score 2) and the lowest in ear position (74% in score 0, 70% in score 1, and 90% in score 2). Sensitivity was the highest in orbital tightening (89% in score 0 and 85% in score 2) and the lowest in nose bulge (19% in score 1). Specificity was the highest in orbital tightening (84% in score 0, 92% in score 1, and 89% in score 2) and the lowest in ear position (77% in score 0, 71% in score 1, and 97% in score 2). The precision and F1 score were also listed in Table 2. The results for score 2 in whisker changes were unavailable because only one image was scored 2. In sum, these results indicate that the predictions of the DeepMGS can approach the ground truth, the scores provided by the experienced human scorer.

### 3.2. Comparison of the Total MGS Score between the DeepMGS and Ground Truth

Linear regression and a Bland–Altman plot (Figure 3) were used for comparing the performance of the DeepMGS in predicting the total MGS score of the ground truth. Linear regression revealed a positive correlation (correlation coefficient (R) = 0.83, *p* < 0.001) between the DeepMGS and ground truth, suggesting that the DeepMGS is comparable to the ground truth. The Bland–Altman plot demonstrated that the 95% limits of agreement (LoA) were −2.96 to 3.412, and most images (92% of the 1504 images) were within the LoA. Specifically, only 35 images (2% of total images) were higher and 88 images (6% of total images) were lower than the LoA. Furthermore, from the Bland–Altman plot, we did not observe systemic bias between the two measurements (Pearson’s correlation, R = −0.095, *p* = 0.1), again supporting the favorable performance of the DeepMGS.

### 3.3. DeepMGS Accurately Classifies Pain and No Pain Conditions

To examine the degree to which the DeepMGS can detect pain in NTG-treated mice, we analyzed its performance in distinguishing images between NTG and saline conditions, which correspond to pain and no pain conditions, respectively (Table 3). In distinguishing these conditions, the DeepMGS had an accuracy of 63% (95% CI: 0.57–0.69, *p* < 0.001), a sensitivity of 62% (95% CI: 0.53–0.71, *p* < 0.001), a specificity of 64% (95% CI: 0.55–0.72, *p* < 0.001), a precision of 58% (95% CI: 0.54–0.62, *p* < 0.001), and an F1 score of 60% (95% CI: 0.57–0.63, *p* < 0.001). For the same classification task, the ground truth had an accuracy of 63% (95% CI: 0.57–0.69, *p* < 0.001), a sensitivity of 64% (95% CI: 0.55–0.74, *p* < 0.001), a specificity of 63% (95% CI: 0.54–0.70, *p* < 0.001), a precision of 63% (95% CI: 0.59–0.67, *p* < 0.001), and an F1 score of 58% (95% CI: 0.55–0.61, *p* < 0.001). The AUROCs of the DeepMGS and ground truth were 0.64 (95% CI: 0.56–0.69, *p* < 0.001) and 0.64 (95% CI: 55–69, *p* < 0.001), respectively (Figure 4a). This suggests that the performance of the DeepMGS was comparable to that of the ground truth provided by the experienced human scorer. The area under the precision-recall curve of DeepMGS and ground truth were 0.63 (95% CI: 0.49–0.66, *p* < 0.001) and 0.63 (95% CI: 0.48–0.67, *p* < 0.001), respectively (Figure 4b). Both areas under the precision-recall curves are higher than the baseline (0.49), a finding indicating that the performance of DeepMGS is still quite good even for the imbalanced datasets (the saline and NTG conditions). We also analyzed the performance in distinguishing NTG and saline conditions by the three apprentice scorers (human scorers 1, 2, and 3; Table 4). Their AUROCs were 0.53 (95% CI: 0.45–0.59), 0.58 (95% CI: 0.50–0.63), and 0.55 (95% CI: 0.47–0.61), respectively, and their correlation coefficients with the ground truth were 0.65, 0.73, and 0.61, respectively. Both the aforementioned values were significantly lower than those of the DeepMGS (Williams’s one-tailed test, all *p* < 0.001), suggesting that the performance of the DeepMGS was higher than that of apprentice scorers.

### 3.4. Heatmap Visualization

To determine the parts of the image that were attended by the DeepMGS when inferring each MGS action unit score, we utilized the gradient-weighted class activation mapping (Grad-CAM) method to compute the heatmap, a visual explanation of the key areas of an image used by a deep learning model [23]. The red areas represent salient areas that affect score prediction, whereas the blue areas are less focused [24] (sample heat map in Figure 5). According to the results, the DeepMGS focused on the appropriate facial area when predicting each of the five action unit scores, a property that further supports its validity.

## 4. Discussion

In this study, we developed the DeepMGS, which applies deep machine learning methods to automatically yield MGS scores with high accuracy. These scores were comparable to those of the experienced human scorer. First, the DeepMGS had high accuracy, 70–95%, in scoring the five MGS action units in migraine-like facial expressions of pain in mice receiving repeated NTG treatment. Second, the total MGS score obtained from the DeepMGS was highly correlated with that obtained from an experienced human scorer, suggesting that the DeepMGS is highly accurate. Third, when distinguishing animals with NTG injection-induced migraine-like pain using the total MGS score, the DeepMGS exhibited performance comparable to that of the experienced human scorer, a finding supporting the high specificity of the DeepMGS. Fourth, the validity of the DeepMGS in each of the five action units can be confirmed by the salience map generated by Grad-CAM. Although the ground truth of MGS action unit scores can be provided by human scorers, the DeepMGS may outperform human scorers in detecting spontaneous pain, most likely because the deep learning method could focus on minute differences that might not be caught by human visual inspection.

Evoked nociceptive responses induced by thermal (such as the hot-plate test) or mechanical (such as the von Frey test) stimulation are commonly applied to investigate pain-induced behaviors in animal models [25,26]. However, spontaneous pain responses, such as the degree and the number of headache attacks in migraine, are difficult to quantitively assess in animals. The MGS was thus developed for the assessment of painful facial expressions as spontaneous painful responses in mice [14]. In a previous work, we utilized the MGS in a mouse migraine model induced by repeated intermittent NTG injections [22]. In this model, both paw allodynia [27] and orbital allodynia [28,29,30] were found due to TGVS activation. This model can be employed as a platform for developing abortive and preventive treatments for migraine since mechanical allodynic responses induced by acute and chronic NTG treatments are sensitive to sumatriptan and topiramate, the abortive and preventive medicines of migraine, respectively [31]. The symptoms of repeated NTG administrations in rodent models align with recurrent episodes of migraine in humans, by a presentation of high MGS scores, paw allodynia, decreased activity, and photophobia [11,22,32]. However, unlike paw allodynia, facial painful expressions cannot be easily quantitated. Besides, multiple NTG-induced migraine episodes substantially increase the number of mouse facial images and thus raise the labor and time costs of evaluating MGS scores. Therefore, the application of deep learning techniques that automatically analyze the animals’ facial expressions could facilitate the research of migraine.

In 2018, Tuttle et al. [20] first developed an automatic deep learning-based scoring method for detecting facial painful expressions of pain in mice. Their method detects pain in a binary manner and mainly focuses on the total MGS score. In contrast, our approach automatically scores each MGS action unit according to the definition of the scale. Additionally, their study focused on high-confidence images and eliminated ambiguous images. Our DeepMGS model analyzes all the images and can predict if the pain exists or not by analyzing the scores of five action units. Thus, our approach involves the scoring of five action units and can thus offer more facial information about the animal. The heatmap results showing that the salient areas of the facial images of mice are comparable to the scores in each action unit further support that the performance of our model is truly based on the corresponding facial areas, except for the whisker change action unit. In the latter action unit, the dataset employed contained only one image with a score of 2 in this action unit, restricting further statistical calculation. Thus, to achieve a high-quality analysis of the whisker change, high-resolution images with proper lightness and contrast are required so that the whisker change can be easily identified.

Following its development, the MGS has been applied to evaluate postoperative pain responses in mice [33,34] and later was also successfully applied to evaluate spontaneous pain responses in rodents [20,33]. The MGS score can also reflect the degree of inflammatory pain [19]. However, whether a decrease in facial grimaces directly indicates a relief of spontaneous pain is being debated [19]. According to a previous study, mice may instinctively control their facial expressions, masking pain to avoid predation [33]. Humans sometimes also suppress facial grimaces when experiencing chronic pain but are unable to completely suppress it as it is not entirely voluntary [35,36]. Therefore, a mouse without a facial grimace may not absolutely indicate that it is free from spontaneous pain. Further supportive measurements, such as unusual body posture, abnormal behaviors, and restless movement, may need to be developed for a comprehensive assessment of spontaneous pain.

Besides the evaluation of facial expressions, the objective analysis of pain behaviors in rodents has been developed in previous studies. For example, by analyzing a video of rats, the animal’s posture and frequency of the activities relating to pain can be measured [37,38]. A study identified pain behaviors by tracking the change in the electromagnetic field generated by the magnets implanted in the rat’s limbs and observed asymmetric limb movements caused by pain [39]. It remains to be elucidated that deep learning models that analyze these videos can also have the potential to quantify these pain behaviors.

The present study has several limitations. First, the current sample size was relatively small; data can be collected from different laboratories to enlarge the sample and ensure generalization. Second, a previous study utilized the Rodent Face Finder to automatically capture and crop images by detecting the eyes and ears of rodents. From these cropped local images, the authors then manually removed low-quality images, such as blurred images, to reduce the manual scoring load and to improve result quality [19]. Because the DeepMGS was trained with random-ordered images, whether the dynamic change in pain over the time course can be captured by this model remains unclear. Third, some problems inherent to the MGS remain in the DeepMGS. For example, the viewing angle of the mouse face affects MGS scoring in some action units, such as the ear rearing up position. Mice tend to pull their ears backward under pain conditions [14], which may be easier to observe in the side view than in the front view. This problem cannot be overcome using the DeepMGS.

In addition to migraine-like facial expressions of pain, whether the DeepMGS can be used to assess spontaneous pain responses in other trigeminal-related pain models, such as dental pulp injury-induced orofacial pain, or chronic pain models, such as fibromyalgia, should be validated. Moreover, whether the DeepMGS can be employed for medical and economic applications, instead of only in laboratories, requires exploration. As the facial expression scale of pain has been validated in humans and rodents [14,19,35], the DeepMGS can provide real-time pain monitoring. This is crucial because the accuracy of the MGS in real-time pain assessment has been challenged considering that it might be lower than that of the retrospective analysis of recorded images by human scorers [40].

This study demonstrated that the DeepMGS exhibited a favorable performance in scoring the five action unit scores and the total MGS score as compared with the experienced human scorer. In addition, the ability of the DeepMGS to classify NTG and saline conditions was comparable to that of an experienced human scorer. Furthermore, compared with the three apprentice scorers, the DeepMGS exhibited a higher AUROC in distinguishing NTG from saline conditions and a higher coefficient of correlation with the experienced scorer, suggesting that the performance of the DeepMGS is superior to that of inexperienced human scorers. Given that the DeepMGS is not prone to human bias and does not involve human labor, its promising performance in distinguishing pain and no pain conditions suggests its future applications to provide real-time and long-term pain monitoring. For example, spontaneous pain, such as migraine, could then be monitored using DeepMGS-like algorithms. Furthermore, monitoring laboratory or economic animals could dramatically improve animal welfare and economic production.

## Figures and Tables

**Figure 1 jpm-12-00851-f001:**
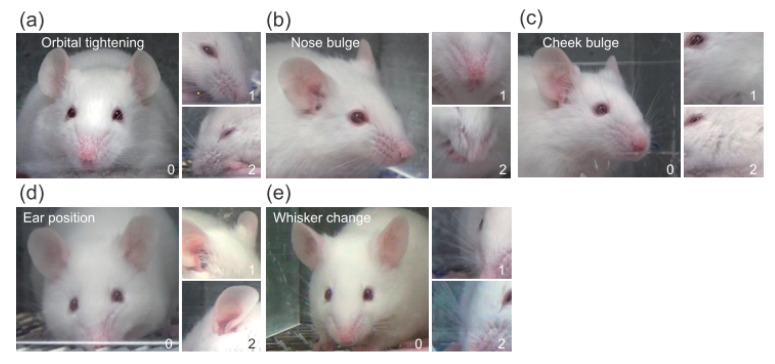
Five MGS action units and the scoring system. Each of the five action units, namely orbital tightening (**a**), nose bulge (**b**), cheek bulge (**c**), ear position (**d**), and whisker change (**e**), were scored using a three-level scale (i.e., 0, 1, or 2). A higher score suggests the scorer had stronger confidence in observing the painful facial expression of mice.

**Figure 2 jpm-12-00851-f002:**
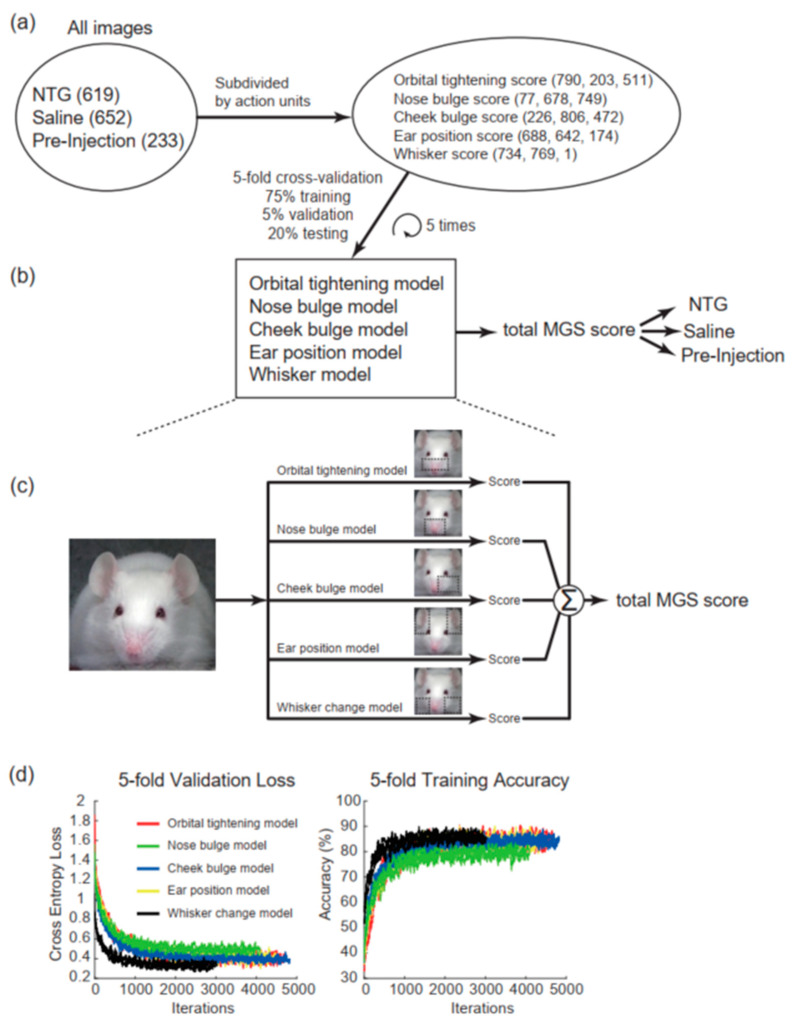
Preparation of the dataset and the architecture of the DeepMGS. (**a**) All images collected from NTG, saline, and preinjection conditions (values in brackets indicate numbers of images) were annotated by the experienced human scorer and subdivided into five MGS action units. Each image was scored using the MGS by the experienced scorer, and the results were used as the ground truth (values in brackets indicate numbers of images annotated with scores of 0, 1, and 2). (**b**) Among the full dataset, 1127 images (75%) were assigned as the training set, 76 images (5%) as the validation set, and 301 images (20%) as the testing set. The five-fold cross-validation process of model training was repeated five times, yielding the estimates of the model’s predictive performance. The DeepMGS contains five classification models to predict the five action unit scores. (**c**) The predicted action unit scores were summed to yield the predicted total MGS score. (**d**) The five-fold validation loss (**left** panel) and training accuracy (**right** panel). The *x*-axes represent the iteration number, and the *y*-axes in the two panels represent the cross-entropy loss and training accuracy, respectively. The five action units are presented with different colors in the diagram.

**Figure 3 jpm-12-00851-f003:**
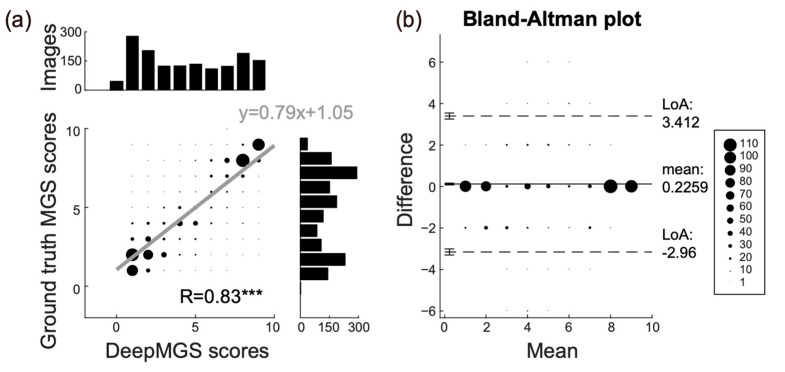
Comparison of the performance of the DeepMGS with the ground truth in the total MGS score prediction. (**a**) The linear regression analysis shows a high correlation coefficient (R = 0.83, *p* < 0.001) in the performance between DeepMGS and ground truth. The *x*-axis of linear regression represents the total MGS score predicted by the DeepMGS, and the *y*-axis represents the ground truth. The black bars represent the number of images with a total MGS score from 0 to 10. The size of dots represents the number of images in each data point. The three asterisks represent *p* < 0.001. (**b**) The Bland–Altman analysis indicates that 93% of the images are within the limits of agreement, which range from −2.96 to 3.412. This result supports a favorable performance of DeepMGS.

**Figure 4 jpm-12-00851-f004:**
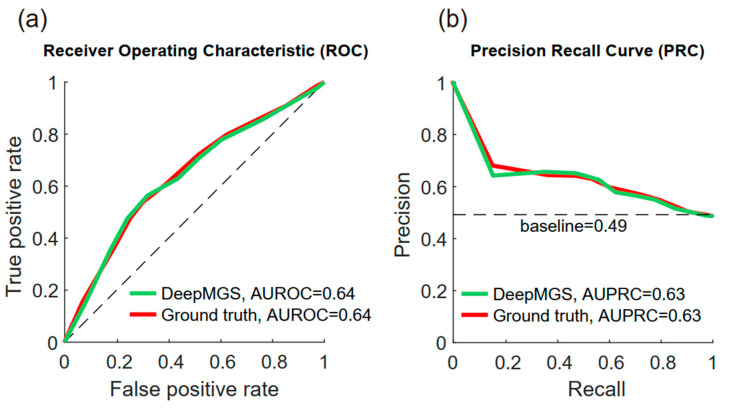
Comparison of the performance of DeepMGS and the ground truth in distinguishing NTG and saline conditions by the receiver operating characteristic (ROC) and precision-recall (PRC) curve analyses. (**a**) The ROC curves of the DeepMGS (green line) and ground truth (red line). The area under the ROCs of the DeepMGS and ground truth were 0.64 and 0.64, respectively, indicating that the performance of the DeepMGS in distinguishing NTG and saline conditions is comparable to that of an experienced scorer. (**b**) The PRC curves of DeepMGS (green line) and ground truth (red line). The area under the PRCs of the DeepMGS and ground truth were 0.63 and 0.63, both higher than the baseline (0.49), suggesting the performance of the DeepMGS is favorable even for the imbalanced datasets.

**Figure 5 jpm-12-00851-f005:**
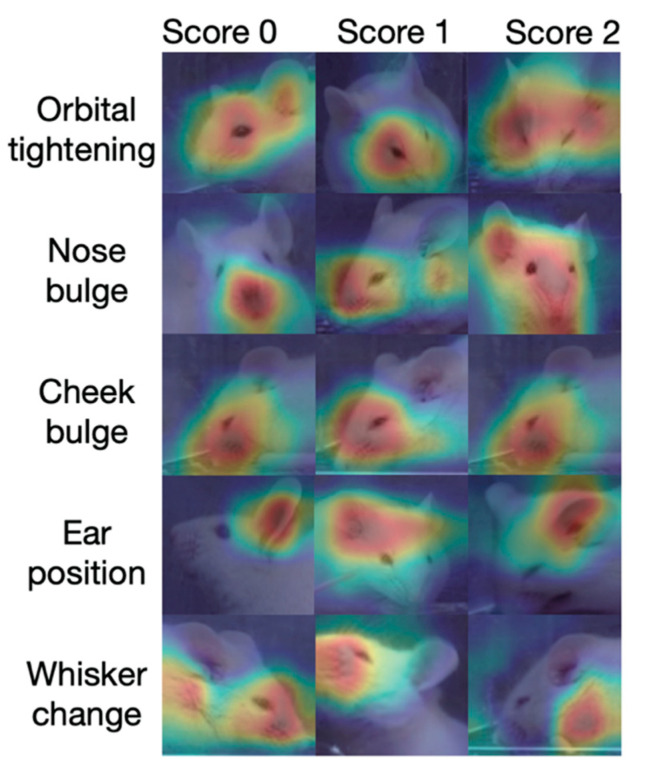
Heatmaps generated using the Grad-CAM method in the mouse facial images. Each image was obtained randomly from the dataset. The red areas represent salient areas that DeepMGS rely on to infer each MGS action unit score. The heatmaps reveal that DeepMGS focuses on the appropriate facial areas when scoring corresponding action units.

**Table 1 jpm-12-00851-t001:** The numbers of images taken of preinjection, saline, and NTG conditions, respectively, used as the training, validation, and testing sets.

Number of Images	Preinjection	Saline	NTG	Total (%)
Training set	174	489	464	1127 (75%)
Validation set	12	33	31	76 (5%)
Testing set	47	130	124	301 (20%)

Images of the preinjection condition were taken before injection in both saline and NTG groups of mice. Images in the training set were used to train the DeepMGS model. Five-fold cross-validation was performed and the process of model training was repeated five times with each of the five subsets. Images in the validation and testing sets were used to validate the performance of this model during training and after training, respectively. Abbreviation: NTG, nitroglycerin.

**Table 2 jpm-12-00851-t002:** DeepMGS performance in the three levels (0, 1, and 2) of each MGS action unit.

Score	Accuracy (95% CI)	Sensitivity (95% CI)	Specificity (95% CI)	Precision (95% CI)	F1 Score (95% CI)
Orbital tightening					
0	0.86 (0.85, 0.88)	0.89 (0.86, 0.91)	0.83 (0.81, 0.86)	0.85 (0.82, 0.87)	0.87 (0.85, 0.88)
1	0.81 (0.79, 0.83)	0.36 (0.30, 0.42)	0.92 (0.90, 0.93)	0.51 (0.44, 0.58)	0.42 (0.38, 0.46)
2	0.88 (0.86, 0.89)	0.85 (0.81, 0.88)	0.89 (0.87, 0.91)	0.77 (0.73, 0.81)	0.81 (0.78, 0.83)
Nose bulge					
0	0.88 (0.86, 0.90)	0.19 (0.13, 0.25)	0.97 (0.95, 0.97)	0.39 (0.28, 0.51)	0.25 (0.20, 0.31)
1	0.74 (0.71, 0.76)	0.74 (0.70, 0.78)	0.73 (0.70, 0.76)	0.64 (0.60, 0.67)	0.68 (0.66, 0.71)
2	0.81 (0.79, 0.83)	0.81 (0.78, 0.83)	0.82 (0.79, 0.84)	0.82 (0.79, 0.85)	0.81 (0.79, 0.83)
Cheek bulge					
0	0.85 (0.83, 0.86)	0.49 (0.44, 0.55)	0.95 (0.94, 0.96)	0.74 (0.68, 0.79)	0.59 (0.55, 0.63)
1	0.73 (0.70, 0.75)	0.81 (0.77, 0.84)	0.67 (0.63, 0.70)	0.65 (0.62, 0.69)	0.72 (0.70, 0.75)
2	0.87 (0.85, 0.88)	0.76 (0.72, 0.79)	0.92 (0.90, 0.94)	0.83 (0.79, 0.86)	0.79 (0.76, 0.82)
Ear position					
0	0.74 (0.71, 0.76)	0.70 (0.67, 0.73)	0.77 (0.74, 0.80)	0.74 (0.71, 0.77)	0.72 (0.70, 0.74)
1	0.70 (0.67, 0.72)	0.68 (0.64, 0.72)	0.71 (0.68, 0.74)	0.55 (0.51, 0.59)	0.61 (0.58, 0.64)
2	0.90 (0.88, 0.91)	0.54 (0.47, 0.60)	0.97 (0.96, 0.98)	0.79 (0.72, 0.85)	0.64 (0.59, 0.68)
Whisker change					
0	0.82 (0.80, 0.84)	0.80 (0.77, 0.83)	0.85 (0.82, 0.88)	0.85 (0.83, 0.88)	0.83 (0.81, 0.84)
1	0.82 (0.80, 0.84)	0.80 (0.77, 0.83)	0.85 (0.82, 0.88)	0.85 (0.83, 0.88)	0.83 (0.81, 0.84)
2 *	-	-	-	-	-

* Data are unavailable as only one image was scored 2 in the “whisker change” action unit. We used the F1 score, the weighted average of precision and recall, to measure the performance of the DeepMGS. Abbreviation: CI, confidence interval.

**Table 3 jpm-12-00851-t003:** Comparison of the performance between the DeepMGS and ground truth in distinguishing pain and no pain conditions.

	Accuracy (95% CI)	Sensitivity (95% CI)	Specificity (95% CI)	Precision (95% CI)	F1 Score (95% CI)
Ground truth	0.63 (0.57, 0.69)	0.64 (0.55, 0.74)	0.63 (0.54, 0.70)	0.63 (0.59, 0.67)	0.58 (0.55, 0.61)
DeepMGS	0.63 (0.57, 0.69)	0.62 (0.53, 0.71)	0.64 (0.55, 0.72)	0.58 (0.54, 0.62)	0.60 (0.57, 0.63)

The images collected in the NTG and saline conditions were considered as pain and no pain conditions, respectively. Abbreviation: CI, confidence interval.

**Table 4 jpm-12-00851-t004:** Comparison of the performance of DeepMGS and three apprentice human scorers in distinguishing NTG and saline conditions with ground truth by linear regression analyses.

	AUROC (95% CI)	Correlation Coefficient with Ground Truth
Ground truth	0.64 (0.55, 0.69)	-
DeepMGS	0.64 (0.56, 0.69)	0.83
Human scorer 1	0.53 (0.45, 0.59)	0.65
Human scorer 2	0.58 (0.50, 0.63)	0.73
Human scorer 3	0.55 (0.47, 0.61)	0.61

Correlation coefficients were estimated by linear regression and revealed positive correlations with ground truth. Human scorer 1, 2, and 3 represent three independent apprentice human scorers. Abbreviations: AUROC, area under the receiver operating characteristic curve; CI, confidence interval.

## Data Availability

Data presented in this study are available upon request from the corresponding author.

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
