# Peer review of "Deep Learning-Based Grimace Scoring Is Comparable to Human Scoring in a Mouse Migraine Model"

_jpm, 2022, doi:10.3390/jpm12060851_

Round 1
Reviewer 1 Report
The authors present a deep learning-based mouse grimace scale (MGS) scoring model which can perform better than human scoring. The proposed DeepMGS estimates the MGS score from the facial expressions of mice. My concerns on the proposed work are given below:
- The article does not give the answer to ‘Deep Learning-Based Grimace Scoring is Superior to Human Scoring in a Mouse Migraine Model’ which is the title of the article. The dataset used for training and validation is labeled by humans and claiming that a deep learning model trained on manually labeled data can predict better than human scorers is totally incorrect.
- Authors should remove all of the commercial details from the manuscript e.g., Lab diet 5001 …, HDR-AXP55 Handy cam…, Millisrol injection, 101 Nippon Kayaku, Tokyo, Japan
- The literature review is very weak as there are only two papers cited from 2021, 2020, and no article is cited from 2019 or 2022. The literature review must be extensively enhanced and it should include different available datasets, existing ML, and/or deep learning-based techniques related to the proposed work.
- How many mice were recorded in the data collection sessions and whether all of the mice belong to the same species? Similarly, whether all of the mice were of the same age, weight, etc.? This is important from the perspective of a deep learning algorithm.
- Details of the deep learning model are not presented. What type of deep learning model is used and what is the architecture of the proposed deep learning model?
- The classes are highly imbalanced e.g., the preinjection class contains only 15% of the data. In the presence of such highly imbalanced data, the model will be overfitting.
- 1500 images are too little for a deep learning model to do any reasonable predictions.
- An 80/20 split for training and validation is not a good idea. The author should create three buckets of data for training, validation, and testing and should use cross-validation techniques e.g. k-fold CV.
- The loss curve, the precision-recall graph should be included.
- Precision, recall, F1-score should be computed.
Reviewer 2 Report
The Authors must improve the conclusions
Reviewer 3 Report
"Deep Learning–Based Grimace Scoring is Superior to Human Scoring in a Mouse Migraine Model" is an interesting article. The aim was to propose a deep learning model, the DeepMGS, that automatically detects the mouse face and estimates the MGS score. Authors used a mouse migraine model induced by repeated and intermittent injections of nitroglycerin (NTG) and a control group injected with saline. The results shown that the DeepMGS achieved an accuracy of 70%–95% in identifying the five action units of the MGS, and its performance (correlation coefficient = 0.84). It is proposed that the DeepMGS is comparable to the experienced human scorer in classifying pain and no pain conditions (area under ROC for both were 0.64).
There are some issues in the article that need to be addressed.
Introduction
At line 63, in a study by Misra et al. [14]., is Tuttle ? There is an error in that reference, correct it.
In the final part of the Introduction, it is suggested to add and delve into the Deep Learning models that have been used in the automatic detection of facial expressions in humans and in animal models of pain and what their results have been.
Also, make clear what the objectives of the article are.
Material and Methods
In section 2.1. Animals. It is mentioned that the data was taken from a previous study, but it should be made clear what the N was and what its characteristics were.
In the DeepMGS development section, it is suggested to add which Deep Learning model or algorithm was used and what its main characteristics are.
In line 157 add the meaning of SGDM
In section 2.5 Statistics, it is suggested to add confidence intervals and significance levels; they are used in the results but they are not defined or mentioned in this section.
Discussion
Tuttle's work is mentioned but the results obtained are not described or compared with those of the article.
It is suggested to briefly and concisely describe the main conclusions.
References
Carefully review all references, there are inconsistencies in them.
For example, some do not have page numbers (references 9,11, etc). Some references use pp 618-626 (reference 17). Some Journal names do not use capital letters in Journal names (reference 13).
Reviewer 4 Report
I have reviewed this manuscript carefully. Some concerns may be answered.
About authors. I haven't seen the #symbol in any article. I think that this can be clarified in authors contribution.
- I consider that abstract must contain less background and more methods explanation.
- Please correct "pain" definition according IASP.
- Other symtoms related with pain can be motor and sensory impairments.
- Please, provide more information about migraine. I consider that it is the major topic of your manuscript. Readers have to be contextualized.
- Provide more information about MGS, such as for example, reliability.
- The number of references is low.
- Figures and tables are appropiated. Please add legend in figures and tables that is necessary.
Author Response
Please see the attachment,
